# Copy Number Variations in the *MICALL2* and *MOGAT2* Genes Are Associated with Ashidan Yak Growth Traits

**DOI:** 10.3390/ani12202779

**Published:** 2022-10-14

**Authors:** Modian Liu, Chun Huang, Rongfeng Dai, Wenwen Ren, Xinyi Li, Xiaoyun Wu, Xiaoming Ma, Min Chu, Pengjia Bao, Xian Guo, Jie Pei, Lin Xiong, Ping Yan, Chunnian Liang

**Affiliations:** 1Key Laboratory of Yak Breeding Engineering Gansu Province, Lanzhou Institute of Husbandry and Pharmaceutical Science, Chinese Academy of Agricultural Sciences, Lanzhou 730050, China; 2Key Laboratory of Animal Genetics and Breeding on Tibetan Plateau, Ministry of Agriculture and Rural Affairs, Lanzhou 730050, China

**Keywords:** *MICALL2* gene, *MOGAT2* gene, copy number variations (CNVs), Ashidan yak, growth traits, association analysis

## Abstract

**Simple Summary:**

Yaks are among the largest high-altitude mammals in the world, and they are ideally adapted to the harsh environmental conditions of the plateau regions. Yaks are thus central to the lives of herdsmen and other local populations in these high-altitude areas. Copy number variations (CNVs) are an important cause of genomic variation in livestock and identifying advantageous CNVs can aid in livestock breeding efforts. In this study, an association between CNVs in the *MICALL2* and *MOGAT2* genes and Ashidan yak growth traits was confirmed, providing a theoretical foundation for Ashidan yak breeding and meat production efforts.

**Abstract:**

Copy number variations (CNVs) are a result of genomic rearrangement affecting DNA regions over 1 kb in length, and can include inversions, translocations, deletions, and duplications. The molecule interacting with CasL-like protein 2 (*MICALL2*) gene is primarily associated with mitochondrial protein targeting and exhibits predicted stress fiber colocalization. The monoacylglycerol O-acyltransferase 2 (*MOGAT2*) gene encodes an enzyme responsible for catalyzing diacylglycerol synthesis from 2-monoacylglycerol and fatty acyl-CoA. For this study, blood samples were obtained from 315 yaks, and the body weight, body length, withers height, and chest girth of these animals were measured at 6, 12, 18, and 30 months of age. Genomic DNA was harvested from the collected blood samples, and CNVs in these samples were detected by qPCR. The resultant data were compared using ANOVAs, revealing significant associations between *MICALL2* gene CNVs and body weight at 6 months of age (*p* < 0.05), body length and chest girth at 30 months of age (*p* < 0.05), and withers height at 18 months of age (*p* < 0.01) in Ashidan yaks. Similarly, *MOGAT2* CNVs were significantly associated with body weight at 6 and 30 months of age (*p* < 0.05), and with withers height at 18 months of age (*p* < 0.01) in these Ashidan yaks. *MICALL2* and *MOGAT2* gene expression was further analyzed in yak tissue samples, revealing that *MICALL2* was most highly expressed in the adipose tissue, whereas *MOGAT2* was most highly expressed in the lung. These results thus confirmed the relationship between CNVs in the *MICALL2* and *MOGAT2* genes and Ashidan yak growth traits, providing a valuable gene locus that can be leveraged for future marker-assisted yak breeding efforts.

## 1. Introduction

Yaks (*Bos grunniens*) are among the most important livestock animals bred in the Qinghai-Tibet Plateau region of China. Yaks are ideally suited to life in this high-altitude environment, as they exhibit excellent tolerance for cold and food scarcity [1]. Initially established in 2019, Ashidan yaks are the first hornless yak breed. Relative to other breeds, these Ashidan yaks are more docile while remaining strong, with dark coloration. They are also well-suited to increased breeding density in an enclosed setting given their docility and lack of horns. Owing to the harsh climatic conditions of the Qinghai-Tibet Plateau, grazing yaks in this region rely on hay intake for 7 months per year during which the average body weight of these animals drops significantly. However, as Ashidan yaks can be raised in an enclosed environment and provided with supplemental nutrition, their body weight can be maintained at stable levels throughout the year [2]. Studies conducted to date have focused on exploring genetic polymorphisms associated with specific yak traits, thus reflecting the rich genetic diversity of these yak breeds while clarifying the genetic relationships between specific breeds and their wild relatives [3]. In addition to permitting the interrogation of the origins and genetic differentiation of these yak breeds, these studies can provide a robust theoretical foundation for directed breeding efforts, enabling the production of superior animals and animal-derived products through the effective leveraging of these genetic resources.

Copy number variations (CNVs), which include insertions, deletions, and other forms of repetition in genomic DNA sequences over 1 kb in length, are a major source of genetic variation among individuals in a given species. These CNVs can arise due to various forms of genomic rearrangement including translocations, inversions, duplications, or deletions of particular regions of the genome [4]. The resulting genetic polymorphisms often produce distinct phenotypic traits in individuals carrying these CNVs [5]. Since their initial codification as a phenotypically important source of variability in 1936 by the American geneticist Calvin Bridges, CNVs have been leveraged as molecular markers capable to guide animal breeding efforts. Recently, CNVs have been used to facilitate optimized cattle and sheep breeding. For example, *MFN1* gene CNVs have been shown to influence Qinchuan cattle body size [6], while *Fec^B^* CNVs are closely related to fertility traits in sheep [7]. In yaks, *CADM2* CNVs are reportedly correlated with body weight [8], whereas *HPGDS* CNVs are significantly associated with both body weight and body length [9].

Members of the molecules interacting with the CasL (*MICAL*) gene family, including *MICAL* proteins and homologous *MICAL-Like* proteins, play important roles in diverse physiological processes [10]. *MICAL* can directly and specifically regulate actin dynamics, promoting semaphorin-induced signal conversion to promote F-actin instability-induced neuronal rejection [11]. *MICAL-Like2* (*MICALL2*) encodes a protein lacking the N-terminal flavin monooxygenase domain present in *MICAL* proteins that is also referred to as the junctional Rab13-binding (*JRAB*) protein, given that it can serve as a Rab13 effector protein [12]. Through its ability to regulate claudin, E-cadherin, and occludin trafficking to the plasma membrane, *MICALL2* serves as an essential regulator of adherens junction and tight junction assembly [13]. *MICALL2* can additionally directly bind to F-actin in a manner that promotes its bundling and stabilization, while further regulating neurite outgrowth and epithelial cell scattering. This gene may thus play an important role in mammalian skeletal muscle development.

The monoacylglycerol O-acyltransferase 2 (*MOGAT2*) gene encodes an enzyme responsible for catalyzing diacylglycerol synthesis from 2-monoacylglycerol and fatty acyl-CoA. It plays an essential role in the digestion, absorption, and metabolic processing of fat. Mice lacking *MOGAT2* expression are reportedly resistant to obesity and high-fat diet-associated metabolic disorders [14]. In Duroc pigs, *MOGAT2* CNVs have also been reported to impact lipid metabolism [15].

*MICALL2* CNVs have previously been reported to be correlated with the body weight, length, and height of young Nanyang cattle, although this correlative relationship was weaker in adult cattle [16]. *MOGAT2* CNVs have also been annotated in a recent whole-gene resequencing study in yaks [17]. In the present study, a quantitative real-time PCR (qPCR) approach was employed to detect *MICALL2* and *MOGAT2* CNVs in Ashidan yaks and to examine the relationship between these CNVs and yak growth traits. In addition, the expression profiles of *MICALL2* and *MOGAT2* were compared across a range of Ashidan yak tissues.

## 2. Materials and Methods

### 2.1. Sample Collection and Growth Trait Measurement

Blood samples (4 mL) were collected from 315 female yaks from Datong Farm, Qinghai Province, China. All yaks were healthy and raised under identical conditions at an altitude of 3200 m. Four phenotypic traits were measured for each yak in this study (body weight, withers height, body length, chest girth) at 6, 12, 18, and 30 months of age. Measurements were made in accordance with the standard Gilbert method [18]. In addition, samples of heart, liver, spleen, lung, kidney, muscle, and adipose tissue were collected from three 18-month-old male yaks which were kept in the same conditions and at the same age, they were used to assess *MICALL2* and *MOGAT2* expression patterns in these different tissues.

### 2.2. Nucleic Acid Isolation

An Easy Pure Blood Genomic DNA Kit (TransGen Biotech, Beijing, China) was used to isolate genomic DNA from yak blood samples, after which a Nanodrop 2000 spectrophotometer (ThermoFisher Scientific Inc., Waltham, MA, USA) was used to assess DNA quality and concentration values. TRIZOL (Transgen Biotch, Beijing, China) was used to extract RNA from heart, liver, spleen, lung, kidney, muscle, and adipose tissue samples, after which 1.2% agarose gel electrophoresis and a spectrophotometer were used to assess RNA concentrations and quality. A PrimeScript™ Reagent Kit with gDNA Eraser (TaKaRa Bio Inc., Dalian, China) was used to prepare cDNA from 1 μg of total RNA. All DNA was stored at −20 °C.

### 2.3. Primer Design and Validation

The online National Center for Biotechnology Information (NCBI) primer-BLAST tool was used to design primers for analyses of *MICALL*2 and *MOGAT2* CNVs and gene expression (Table 1). Optimal primer temperatures were determined through PCR and 1.5% agarose gel electrophoresis analyses. Each individual reaction included 10 μL of 2× Accurate Taq Master Mix (Accurate Biotechnology, Hunan, China), 0.4 μL of each primer, 0.1 μL of DNA, and 8.2 μL of sterile H_2_O. Thermocycler settings were as follows: 94 °C for 30 s; 30 cycles of 98 °C for 10 s, 55–60 °C for 30 s, and 72 °C for 1 min; 72 °C for 2 min.

### 2.4. MICALL2 and MOGAT2 CNV and Tissue Expression Analyses

For analyses of *MICALL2* and *MOGAT2* CNVs, basic transcription factor 3 (*BTF3*) served as an internal reference gene [19]. Genomic DNA was stained with SYBR^®^ Green Pro Taq HS (Accurate Biology, Hunan, China), after which CNVs were measured for these two target genes via qPCR using a LightCycler^®^ 96 Instrument (Roche, Basel, Switzerland) [20]. Each reaction included 10 μL of SYBR^®^ Green Pro Taq HS, 0.4 μL of each primer, 1 μL of DNA, and 8.2 μL of sterile H_2_O. Thermocycler settings were as follows: 90 °C for 30 s; 45 cycles of 95 °C for 5 s and 60 °C for 30 s; 95 °C for 5 s; 65 °C for 60 s; hold at 95 °C. Glyceraldehyde 3-phosphate dehydrogenase (*GAPDH*) is frequently used as a normalization control in tissue analyses [21], and it was thus used as a reference gene for analyses of *MICALL2* and *MOGAT2* gene expression in individual tissues. All qPCR reactions were performed using the same conditions used for CNV detection. Analyses were repeated in triplicate, and results are reported as means ± standard deviation (SD).

### 2.5. Association Analyses

*MICALL2* and *MOGAT2* CNVs were classified as gains, losses, or normal (non-CNVs) using the $2\times 2^{-\Delta \Delta Ct}$ method, where $\Delta Ct=Ct_{target\;gene}-Ct_{reference\;gene}$ [22]. Relationships between these CNV types and yak growth traits were then analyzed using analyses of variance (ANOVAs, SPSS Version19, IBM, New York, NY, USA). A $2\times 2^{-\Delta \Delta Ct}$ value < 2.5 was classified as loss, while $2\times 2^{-\Delta \Delta Ct}$ = 2.5 was classified as normal, and $2\times 2^{-\Delta \Delta Ct}$ > 2.5 was classified as gain. The following linear model was used to explore the relationship between CNVs and growth traits: $Y_{j}=\mu +CNV_{j}+e_{j}$, where *Y_j_* is the observed growth trait value, *μ* is the overall mean value for that growth trait, *CNV_j_* is the effect of the three CNV types on this phenotype, and *e_j_* is the random residual error.

## 3. Results

### 3.1. Associations between Ashidan Yak CNVs and Growth Traits

*MICALL2* CNV classifications for the 315 sampled Ashidan yaks included 142, 108, and 65 loss-type, normal-type, and gain-type animals, respectively (Table 2). In addition, the *MOGAT2* CNV classifications for 283 sampled Ashidan yaks included 130, 87, and 66 loss-type, normal-type, and gain-type animals, respectively (Figure 1).

Correlation analyses revealed that *MICALL2* and *MOGAT2* CNVs were significantly correlated with certain growth traits in this yak population. Specifically, *MICALL2* CNVs were significantly associated with body weight in 6-month-old yaks (*p* < 0.05), body length in 30-month-old yaks (*p* < 0.05), chest girth in 30-month-old yaks (*p* < 0.05), and withers height in 18-month-old yaks (*p* < 0.01) (Table 2). Similarly, *MOGAT2* CNVs were significantly associated with body weight in 6-month-old and 30-month-old yaks (*p* < 0.05) and with withers height in 18-month-old yaks (*p* < 0.01) (Table 3)

### 3.2. MICALL2 and MOGAT2 Gene Expression Profiles

To validate the relationships between the expression of *MICALL2* and *MOGAT2* and the growth and development of Ashidan yaks, the mRNA expression profiles for these genes were analyzed in samples of Ashidan yak heart, liver, spleen, lung, kidney, muscle, and adipose tissue. These analyses revealed that *MICALL2* was expressed at significantly higher levels in adipose tissue relative to lung or kidney tissue (*p* < 0.05) (Figure 2A). Moreover, *MOGAT2* was expressed at significantly higher levels in lung samples relative to other tissues (*p* < 0.05) (Figure 2B).

## 4. Discussion

In recent years, there have been rapid advances in the field of molecular breeding, with marker-assisted selection (MAS) having accelerated the slower pace of more traditional breeding strategies [23]. In an analysis of 488 Chinese Qinchuan cattle, for example, researchers detected four single nucleotide polymorphisms (SNPs) in the *CFL1* gene that were significantly correlated with key growth traits including body weight, body length, chest width, and chest depth [24]. Chickens exhibit high levels of *NUDT15* expression in the bone tissue, and two insertion-deletion (InDel) variants in the promoter region of this gene have been linked to significant variability in chicken growth and carcass weight [25]. In sheep, *TOP2B* CNVs are also reportedly associated with the body length and chest circumference of sheep such that they may offer value for MAS-based breeding programs [26]. These genetic variations in livestock species can also influence breeding outcomes, with economically important traits such as meat quality and carcass weight being the primary focus of these analyses. CNVs in the *SERPINA3-1* [27], *DYNC1I2* [28], *PLA2G2A* [29], and *SYT11* [30] genes can influence beef cattle growth traits, while CNVs in the *CCSER1* [31], *MYLK4* [32], *PIGY* [33], and *KAT6A* [34] genes can influence sheep and goat growth traits.

Here, Ashidan yaks served as experimental subjects, and CNVs in the *MICALL2* and *MOGAT2* genes were detected by qPCR to determine whether these variations were related to important yak growth traits. In addition, the expression profiles of these two genes were surveyed across a variety of yak tissues, ultimately supporting their potential effects on yak growth. The $2\times 2^{-\Delta \Delta Ct}$ method [35] was used to normalize these data, with results being classified into three CNV categories (loss, gain, or normal). This approach ultimately revealed that *MICALL2* CNVs were significantly correlated with body weight at 6 months of age, body length and chest girth at 30 months of age, and withers height at 18 months of age. Similarly, *MOGAT2* CNVs were significantly associated with body weight at 6 and 30 months of age, and with withers height at 18 months of age in these Ashidan yaks. For the first three traits associated with *MICALL2* CNVs, loss and normal classifications were associated with better phenotypic outcomes relative to gains in these yaks, whereas withers height values were higher for yaks exhibiting *MICALL2* gains. We thus hypothesized that these *MICALL2* CNVs adversely impact weight gain in young yaks, but positively impact the height of mature adult yaks while negatively impacting body length and chest girth. Similarly, *MOGAT2* gains were significantly associated with poorer body weight in 6- and 30-month-old Ashidan yaks but were favorably associated with the withers height of 18-month-old yaks. In general, normal-type and gain-type were better than the loss-type, so the individuals with *MICALL2* gene loss-type could be eliminated in yak breeding, and the growth traits of *MOGAT2* gain-type were also generally worse than those of loss-type and normal-type, these guesses need to be confirmed by subsequent experiments. The two gene results require further experimental validation. Gene expression profiling confirmed that *MICALL2* was expressed at high levels in adipose, lung, and kidney tissues, while *MOGAT2* was highly expressed in lung tissue samples. As such, these genes may govern the growth and development of these target tissues in Ashidan yaks.

Although initially identified in mammals, the *MICAL* protein family has primarily been characterized in studies of *Drosophila* model animals [11]. These *MICAL* family members can reportedly interact with F-actin and influence cytoskeletal development. *MICAL* gene expression has been detected in both embryonic and adult nervous system tissues [36]. High levels of *MICALL2* expression have been reported in a range of malignancies including ovarian, gastric, and breast cancers, consistent with the high levels of variability observed for this gene [37]. This suggests that this gene can readily undergo mutation and that it can affect adipose and nervous system tissue development. As such, *MICALL2* CNVs in yaks may similarly influence adipose tissue development in a manner that ultimately alters the growth traits observed in these yaks.

*MOGAT2* encodes an enzyme responsible for triacylglycerol resynthesis, which is critical to fat absorption in the small intestine. Given its role in the absorption of dietary fat, *MOGAT2* has been established as an important driver of obesity in humans [38]. *MOGAT2* deficiency results in increased energy expenditure and the suppression of weight gain in a genetic mouse model of obesity [39]. In yaks, however, fat intake and storage are both beneficial adaptations to the harsh plateau environment. Two studies have previously identified *MOGAT2* CNVs in yaks and Duroc pics through whole-genome sequencing and annotated the function of this gene [15,17]. Accordingly, the association between *MOGAT2* CNVs and yak growth traits was herein analyzed. In Ashidan yaks, these *MOGAT2* CNVs were associated with several growth traits including body weight and body height, potentially mediating these effects through increased fat absorption and consequent increases in yak growth performance. These CNVs may also influence lung development, thereby impacting blood oxygen content or other factors that have the potential to impact overall yak growth and development [40].

## 5. Conclusions

In summary, these results offer new evidence that CNVs in the *MICALL2* and *MOGAT2* genes are associated with yak growth traits. As such, these findings provide a robust theoretical foundation for the use of these two genes as targets for molecular breeding efforts aimed at improving Ashidan yak growth performance.

## Figures and Tables

**Figure 1 animals-12-02779-f001:**
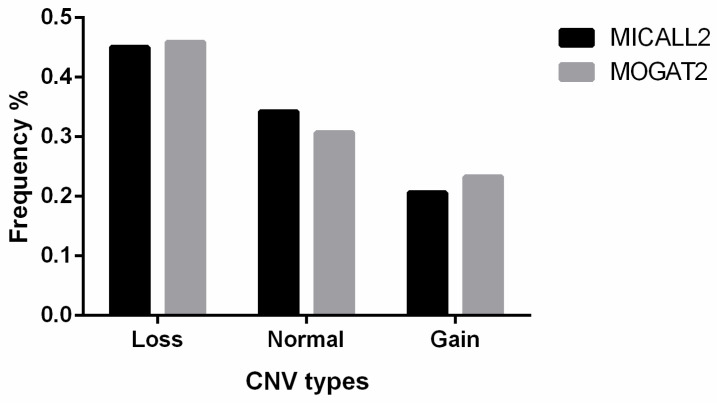
*MICALL2* and *MOGAT2* gene CNV proportions in analyzed Ashidan yaks.

**Figure 2 animals-12-02779-f002:**
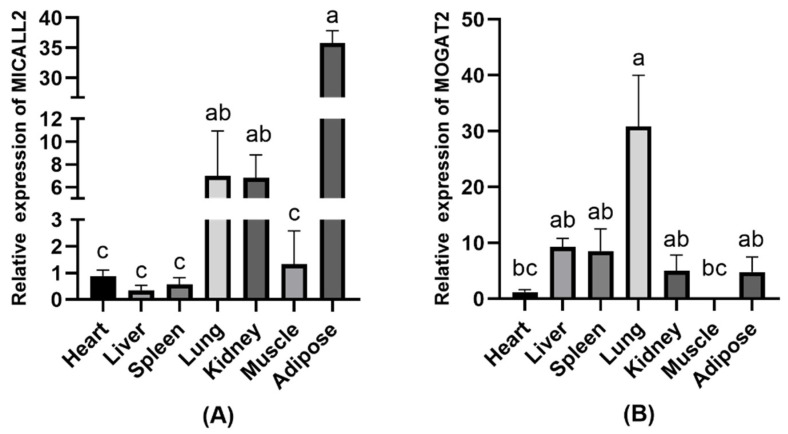
*MICALL2* and *MOGAT2* expression in Ashidan yak tissue samples. (**A**) *MICALL2* and (**B**) *MOGAT2* expression levels in different tissues. a–c denote significant differences at *p* < 0.05.

**Table 1 animals-12-02779-t001:** Primer sequences.

Level	Gene	Primer Pairs Sequence (5′-3′)	Product Length (bp)	Temperature (°C)
DNA	*MICALL2*	F: CCGTCGTCTAATGCCAGTGA R: CATCTTTCCGCTGGACGGTA	133	58.0
	*MOGAT2*	F: CGCTGGTCAAGACTGCCTAT R: ACAGTGAGGAAAACCCGGTG	126	60.0
	*BTF3*	F: AACCAGGAGAAACTCGCCAA R: TTCGGTGAAATGCCCTCTCG	166	60.0
mRNA	*MICALL2*	F: CCTCATGGTGGACTGGTTCC R: CAATGATGTCGCTTCGGCTG	239	59.9
	*MOGAT2*	F: CGCTGGTCAAGACTGCCTAT R: CATCATCAGATGTGGGCGGA	155	59.8
	*GADPH*	F: AATGAAAGGGCCATCACCATC R: GTGGTTCACGCCCATCACA	204	58.8

Note: F: forward primer; R: reverse primer.

**Table 2 animals-12-02779-t002:** Correlations between *MICALL2* CNVs and Ashidan yak growth traits.

Age	Growth Trait	CNV Type (Mean ± SD)			*p*-Value
		Loss (142)	Normal (108)	Gain (65)	
6 months	Body weight (kg)	82.70 ± 9.553 ^c^	86.54 ± 11.289 ^a^	84.10 ± 10.115 ^ab^	0.0151 *
	Withers height (cm)	94.41 ± 4.767	94.69 ± 5.445	93.70 ± 6.338	0.504
	Body length (cm)	90.91 ± 6.425	92.49 ± 7.564	93.03 ± 9.136	0.0971
	Chest girth (cm)	124.46 ± 7.554	123.51 ± 7.693	124.13 ± 8.021	0.626
12 months	Body weight (kg)	82.28 ± 9.819	83.69 ± 11.064	81.61 ± 11.056	0.413
	Withers height (cm)	90.46 ± 4.034	90.52 ± 4.618	90.46 ± 3.942	0.992
	Body length (cm)	95.81 ± 4.085	96.48 ± 5.482	95.73 ± 4.965	0.478
	Chest girth (cm)	117.36 ± 4.848	117.47 ± 5.726	116.49 ± 4.700	0.447
18 months	Body weight (kg)	121.63 ± 13.048	124.49 ± 14.391	120.82 ± 11.663	0.226
	Withers height (cm)	100.42 ± 6.392 ^C^	102.55 ± 6.057 ^AB^	103.59 ± 5.308 ^A^	0.00138 **
	Body length (cm)	101.10 ± 5.742	102.53 ± 6.349	102.20 ± 4.448	0.427
	Chest girth (cm)	137.40 ± 10.263	138.99 ± 10.601	139.01 ± 9.882	0.149
30 months	Body weight (kg)	155.77 ± 14.586	156.62 ± 17.498	154.47 ± 13.713	0.771
	Withers height (cm)	99.26 ± 5.195	100.45 ± 5.126	99.46 ± 4.895	0.288
	Body length (cm)	113.70 ± 5.022 ^a^	113.49 ± 6.260 ^ab^	111.20 ± 5.692 ^c^	0.031 *
	Chest girth (cm)	147.18 ± 7.638 ^ab^	148.32 ± 8.256 ^a^	144.09 ± 10.148 ^c^	0.038 *

Note: Lowercase letters (a–c) denote significant differences (*p* < 0.05), capital letters (A–C) denote highly significant differences (*p* < 0.01); SD: standard deviation; * *p* < 0.05; ** *p* < 0.01.

**Table 3 animals-12-02779-t003:** Correlations between *MOGAT2* CNVs and Ashidan yak growth traits.

Age	Growth Trait	CNV Type (Mean ± SD)			*p*-Value
		Loss (130)	Normal (87)	Gain (66)	
6 months	Body weight (kg)	85.23 ± 11.014 ^ab^	85.51 ± 10.266 ^a^	81.85 ± 9.757 ^c^	0.0462 *
	Withers height (cm)	94.12 ± 5.098	94.34 ± 6.117	94.72 ± 4.958	0.804
	Body length (cm)	91.27 ± 6.768	92.26 ± 7.128	92.78 ± 8.543	0.353
	Chest girth (cm)	124.18 ± 8.503	124.80 ± 6.725	123.00 ± 7.479	0.384
12 months	Body weight (kg)	83.43 ± 10.559	83.98 ± 9.891	80.89 ± 11.325	0.171
	Withers height (cm)	90.43 ± 4.315	90.66 ± 4.324	90.23 ± 4.109	0.833
	Body length (cm)	96.09 ± 4.885	95.94 ± 4.793	95.61 ± 5.574	0.822
	Chest girth (cm)	117.30 ± 5.631	117.42 ± 4.582	117.19 ± 5.279	0.959
18 months	Body weight (kg)	123.76 ± 12.689	123.09 ± 13.356	119.34 ± 15.277	0.162
	Withers height (cm)	100.38 ± 5.891 ^c^	102.20 ± 6.515 ^ab^	103.20 ± 5.381 ^a^	0.00725 **
	Body length (cm)	101.94 ± 5.552	101.60 ± 5.908	101.56 ± 5.781	0.905
	Chest girth (cm)	138.47 ± 10.138	137.80 ± 10.531	138.07 ± 10.473	0.898
30 months	Body weight (kg)	157.26 ± 14.992 ^ab^	158.42 ± 14.525 ^a^	151.16 ± 17.188 ^c^	0.0328 *
	Withers height (cm)	99.38 ± 5.006	99.98 ± 5.234	99.45 ± 4.655	0.738
	Body length (cm)	113.58 ± 5.454	113.63 ± 5.943	112.59 ± 6.130	0.848
	Chest girth (cm)	146.67 ± 8.325	147.44 ± 8.779	147.18 ± 8.215	0.562

Note: Lowercase letters (a–c) denote significant differences (*p* < 0.05); SD: standard deviation; * *p* < 0.05; ** *p* < 0.01.

## Data Availability

The study did not report any data.

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
