# Peer review of "Copy Number Variations in the MICALL2 and MOGAT2 Genes Are Associated with Ashidan Yak Growth Traits"

_animals, 2022, doi:10.3390/ani12202779_

Round 1

Reviewer 1 Report

Q1.  In the introduction, the reasons for selecting MICALL2 gene and related literature are introduced, but the reasons for selecting MOGAT2 gene and related literature are missing.

Q2.  In Association analysis, how to use ANOVA for analysis, please describe the analysis process and how to type the three genotypes?

Q3.  The optimal temperature for PCR detection of primers was used in Primer Design. Please attach the detection reagent and detection conditions.

Q4.  Why BTF3 gene was used as a reference gene for CNV detection? Please explain the reasons and refer to relevant literature?

Q5.  Table 3 is missing comments.

Q6.  The 'A,B' should be bold in the comments in Figure 2.

Q7.  The reference format needs to be modified.

Q8.  In the discussion, the description of results should be appropriately reduced, and the discussion of results and the influence of two genes on growth traits should be increased.

Q9.  The experimental results of expression profile should be discussed in conclusion.

Reviewer 2 Report

The manuscript presents interesting research about Copy Number Variation of the MICALL2 and MOGAT2 genes in the Ashidan yak, and their association with growth traits. Unfortunately, I found some shortcomings and inaccuracies in the manuscript, which must be corrected before considering its publication.

Generally:

There is no information about sex and relations between the animals. This should be clearly described in the “Sample collections and measurement of growth traits” section. The gender and relations between animals are very important factors affecting obtained gene expression levels.

I have some doubts concern the significance obtained for data and presented at Tables. How can you obtain the significance when the SD values is so high is so high that the averages equalize? Please explain.

When talking about the significance don’t use the word “extreme”, results can be highly significant.

Tables 2 and 3 should be corrected. Use the same number of places after the coma. Change “P” to “p” in the first row in both tables.

Punctuation should be checked throughout the manuscript. Commas are missing in many places.

Articles should also be completed. Some examples of the missing articles in the text: the positive (line 24), the stable (line 54), the trafficking (line 85) and many more.

The other inaccuracies and errors in line by line review:

Line 2 – I suggest a change in the title – “gene” to “genes”

Line 22 – “fragment” should be in the plural form

Lines 21 and 24 – a missing verb

Line 24 – “to” is unnecessary

Line 85 – “through” should be replaced with “by”

Line 99 – “gene” should be in the plural form

Lines 123 and 133 – a missing verb

Line 142 – “gene” should be in the plural form

Line 159 – which non-parametric test was used?

Line 174 – “gene was” should be in the plural form

Line 176 – “was” should be in the plural form

Line 213 – “concerned” should be replaced with “concerning”

Line 263 – no space after the parentheses

Line 265 – rewrite the gene with italics

In my opinion, the manuscript presents interesting research but it requires corrections before considering its publication. I suggest a major revision.

Reviewer 3 Report

General Comments:

The manuscript is well structured and clear. Minor revisions to language and grammar will further improve the manuscript.

Additionally the use of words/phrases such as “we, we took, generally speaking”; should be avoided for scientific publications. I suggest you reduce the use of active voice

The manuscript employed a scientifically sound experimental design to test the association of CNVs to growth traits.

However, this study is similar to Ge et al 2019 reference [6] and Huang et al 2021 reference [7].  In the methodology there is direct copying of sentences from Huang et al 2021, if the same approach was used, it would be advisable to cite the source and paraphrase correctly.

Figure 1, may seem biased, as the population size for both genes used were different during the comparison. (MICALL2: n=315, MOGAT2: n =283). All other tables and figures have been correctly labeled and are easy to understand.

The results are clearly shown and represented well by figures and tables. Whereas the discussion lacks some minor additional referencing for statements made.  The conclusions drawn from this study are captured clearly, collectively summarizing the main output, which is the verification of the association of the two genes to growth traits in the Ashidan yak.

Regarding the ethics, the collection of tissue samples is not included. As it stands now, it seems as if tissue sample collection was not approved by the Lanzhou Institute of Husbandry and Pharmaceutical Science of CAAS.

Specific Comments:

Line 15: “can” should be replaced with “are”

Line 27: “We took” should be replaced with “Blood was collected…”

Line 28: …Chest girth at the age of 6, 12, 18-28 and 30 months. Try to correct the rest of the manuscript accordingly

Line 30-34: This sentence is too long. It should be split into two.

Rephrase “extremely significant correlation” What does this even mean at p<0.01?

Line 35: Do not use “we”. Rephrase the sentence – “The expression of MICALL2 and MOGAT2 was detected in the yak tissues.”

Line 37: Remove “the highest” and replace with “highly”

Line 47: Remove “better”

Line 54-58: Multiple ideas in one sentence. Separate and support each with a reference.

Line 58: What does refer to in the sentence: “Furthermore, it can…” Rephrase, for more clarity.

Line 59: The term “new strain” does not seem to make sense in the context of the sentence.

Line 64-65: This sentence is repeated from the abstract line 22-23.

Lind 70: Provide the reference at the end of the sentence to support your statement of “…Calvin Bridges…”

Line 90-92: Add reference for each of those sentences.

Line 107: The collection of blood is mentioned here, what about the tissues mentioned in lines 117-118.

Line 112-113: This copied directly from Huang et al 2021 ref 7.

Line 127-130: This copied directly from Huang et al 2021 ref 7.

Line 134: The optimum temperature were determined for each primer, but it is not shown in Table 1.

Line 169-172: What was the reason for using 283 yaks instead of the 315 yaks for MOGAT2 CNV detection? Because the graph in figure 1 showing a higher loss in MOGAT2 than MICAL2L, is this not biased towards MOGAT2 as it has a smaller population size.

Line 179: Remove comma and replace with “and” before the word “withers”

Line 189: Remove “we also”. Rephrase the sentence: “…,the mRNA expression levels were detected for the two genes in several tissues (heart, liver…) .”

Line 215-217: Remove the word “gene” after each of the mentioned genes.

Line 244: After the last comma in line 244, add “whereas”

Line 249: Provide reference for the sentence ending in “…studied in drosophila as experimental animals.” And provide references for “According to relevant studies…”

Line 272: Provide supporting reference for “… affect biochemical indexes such as blood oxygen content….”

Line 292: Again the mention of tissue sample collection was not mentioned here. 

Round 2

Reviewer 2 Report

The manuscript presents interesting research about Copy Number Variation of the MICALL2 and MOGAT2 genes in the Ashidan yak, and their association with growth traits. After the first reviews, the vast majority of errors in the manuscript were eliminated. Unfortunately, I still found some shortcomings, which must be corrected before considering its publication. Please refer to the points below.

There is no information about the relations between the animals. This should be clearly described in the “Sample collections and measurement of growth traits” section. The relations between animals can be an important factor affecting obtained gene expression levels.

Line 145 – no spaces in front of the parentheses

Line 246  – “drosophila” should be italicized

Author Response

Point 1: There is no information about the relations between the animals. This should be clearly described in the “Sample collections and measurement of growth traits” section. The relations between animals can be an important factor affecting obtained gene expression levels.

Response 1: The “Sample collections and measurement of growth traits” were supplemented with information about the relationship between animals and the gene expression section.

Point 2: Line 145 – no spaces in front of the parentheses

Response 2: The space before the parentheses has been added.

Point 3: Line 246  – “drosophila” should be italicized

Response 3: The word “drosophila” has been italicized.